# Differences in the Potential Accessibility of Home-Based Healthcare Services among Different Groups of Older Adults: A Case from Shaanxi Province, China

**DOI:** 10.3390/healthcare8040452

**Published:** 2020-11-01

**Authors:** Lijian Wang, Xiaodong Di, Liu Yang, Xiuliang Dai

**Affiliations:** School of Public Policy and Administration, Xi’an Jiaotong University, No 28 Xianning West Road, Xi’an 710049, China; wanglijian2@mail.xjtu.edu.cn (L.W.); yangliu777@stu.xjtu.edu.cn (L.Y.); xiuliangdai@stu.xjtu.edu.cn (X.D.)

**Keywords:** potential accessibility, healthcare service, perceived vulnerability, difference analysis

## Abstract

With the increase of the aging population and the lack of family care, home-based healthcare services have gradually become the main model to cope with aging, so local governments have invested heavily in the construction of home-based healthcare services. However, healthcare services still have problems such as low resource utilization and imbalanced development. The reason is that the supply and demand of healthcare services are not matched and the potential accessibility is low. Therefore, based on the supply and demand of healthcare services, this article pulls out the spatial and social factors that affect the potential accessibility, and tests the influence of individual factors on the potential accessibility among different groups of older adults. It is found that the perceived vulnerability of the older adults will reduce the potential accessibility of healthcare services. The psychosocial status, income and education level with the willingness to use healthcare services of the older adults are directly proportional, while residence has a negative impact on the potential accessibility. Finally, based on this finding, this article puts forward feasible suggestions from the perspective of policy content, publicity, and implementation.

## 1. Introduction

In China, with the rapid increase of the aging population and the weakening of the family care function, home-based healthcare services have become the main way to cope with aging [1,2]. Home-based healthcare services mean that older adults live at home and the healthcare services centers in community provide daily care, such as medical care, housekeeping, entertainment and mental care [3]. In 2019, the “Opinions of the General Office of the State Council on Promoting the Development of Healthcare Service for the Older Adults” (No. 5) emphasized that it was necessary to develop a healthcare service systems based on the home, supported by the community, supplemented by the institutions and combined with medical care. Therefore, local governments are also actively responding to the policy and vigorously developing home-based healthcare services. By the end of 2019, there were 204,000 healthcare institutions and facilities for the older adults, 7.75 million healthcare beds, with an increase of 6.6% over the previous year, and 30.5 healthcare beds per 1000 older adults in China. Among them, there were 64,000 healthcare institutions and facilities, 101,000 mutual-aid facilities for the older adults, and a total of 3.362 million beds in the community. However, the development of home-based healthcare services still has some problems of the inactive market, the imbalanced development [4], low resource utilization and poor service quality. The reason is that the supply and demand of healthcare services is out of balance and older consumers have low opportunities and willingness to use healthcare services. In order to realize rapidly the equal and high-quality development of healthcare services, we must improve the accessibility of healthcare services and coordinate the supply and demand of healthcare services.

Regarding the research on the accessibility of healthcare services, most scholars only focus on the accessibility of healthcare services facilities for the older adults [5,6,7]. Although some scholars have also paid attention to the accessibility of intelligent services [8], network service [9], public service [10], life satisfaction [11], etc., all are based on the realized accessibility, ignoring the potential opportunities and willingness of the older adults to use healthcare services. This not only hinders the access to healthcare services, but also is not conducive to stimulate the healthcare service market. In addition, the older adults with different characteristics have different willingness to use healthcare services, and their potential accessibility will be different, so the allocation of healthcare service resources needs to be adapted to local conditions. Therefore, it is necessary to study the factors affecting the potential accessibility and the reasons causing the differences.

On the study of factors affecting potential accessibility, potential accessibility is the opportunity and willingness of users to the service system, and involves geographical factors and social factors [12]. Geographical factors mainly refer to the distance between users and the service system, and social factors include factors that affect the willingness of using healthcare services, like economic [13], quality [14], information [15] and other factors. Regarding the study on the potential accessibility of healthcare services, Hartlaub et al. studied the potential accessibility of mental health services and found that there were many factors affecting the likelihood of accessing mental health services. These factors included the inability to pay for services, the lack of transportation and the negative attitudes [16]. Okpala studied the potential accessibility of medical services and found that collaborative leadership positively influences access to quality healthcare services [17]. Therefore, in terms of the potential accessibility of healthcare services, we should not only pay attention to spatial factors, but also social factors that affect healthcare services.

On the study of differences of potential accessibility, some scholars believe that poor access to healthcare may be as a result of other socioeconomic conditions such as poverty and poor education, which increases the vulnerability of an older adult and likely association with depressive symptoms [18,19]. Some else believe that urban-rural differences have caused differences in the potential accessibility of healthcare services. Gao studied maternal health service utilization and pointed out that in urban regions, pockets of maternal health disparities exist despite close distance to facilities and standard quality of care. In rural regions, there were areas with long distances to facilities and low quality of care, resulting in poor maternal service usage [20]. In addition, there are individual factors such as age, gender, and income that cause differences on potential accessibility [21].

Therefore, this article constructs an analysis framework for the potential accessibility based on the demand and supply of healthcare services. Then, it explores the different impact of the geographic distance and perceived vulnerability on the potential accessibility among different groups of older adults. Finally, it analyzes the reasons of differences on potential accessibility from the perspectives of four individual characteristics of education, income, residence, and psychosocial status, and puts forward some suggestions.

## 2. Analysis Framework

The development of healthcare services depends on two aspects of supply and demand. From the perspective of supply, the allocation of healthcare service resources determines the spatial distribution of healthcare service supply, as well as the geographical opportunities for the older adults to access and use healthcare services. From the perspective of demand, the demand of healthcare services depends on the older adults’ perceived ability to healthcare services, not only the external perceived ability to access healthcare services system, but also the internal perceived ability of willingness to use healthcare services [22]. Geographic distance as a spatial factor and perceived ability as a social factor together affect the accessibility of healthcare services. In terms of spatial factor, the closer the older adults are to the healthcare services supply center, the greater the opportunity is to access and use the healthcare services. However, in terms of perceived ability, the older adults are generally vulnerable to the perception of healthcare services due to the decline in physical function, cognitive ability, and information acquisition ability [23], which hinders consumers’ willingness to find and approach to healthcare services [24]. Ribeiro et al. pointed out that the factors that might interfere in the older adults’ perceived health included age, sex, chronic health conditions, family and social support, functionality, and lifestyle [25]. Pergher found the positive correlations between age, sex, and education with gray matter volume, which supported that age, sex, and education affected individual cognitive, neurophysiological and structural characteristics [26].

Therefore, this article draws on the selection mechanism of individual characteristics from Myall [23] and Keylla [27] and divides the older adults into four groups from individual characteristics of psychosocial status, income, residence, and education. Then, it analyzes the different impact of the distance to service and perceived vulnerability on potential accessibility by comparison within the group. The analysis framework and hypotheses (H) are shown in Figure 1.

If the older adults have a strong perception to the healthcare services system, the older adults are more likely to use healthcare services, and the enjoyment of healthcare services in turn reduces the perceived vulnerability of the older adults. On the contrary, the inaccessibility of healthcare services due to certain obstacles or lack of services will also increase the perceived vulnerability of the older adults [28]. In addition, under the same circumstances, the perceived vulnerability of the different older adults is different. Loss of independence, mental decline, physical decline, social isolation and financial threats and so on are all important factors affecting the perceived vulnerability of the older adults. So in terms of internal perceived ability, the older adults have different judgments about whether they need healthcare services. In terms of external perceived ability, individuals have different abilities to perceive, find, and approach healthcare services. When the older adults use healthcare services, the perceived vulnerability will weaken the satisfaction [29], reduce the frequency of using healthcare services and the interaction with healthcare personnel [30]. The older adults will miss some healthcare services that are beneficial to them, thereby reducing the potential accessibility of healthcare services.

**Hypothesis** **1** **(H1).**
*The perceived vulnerability of the older adults reduces the potential accessibility of healthcare services.*


From the perspective of the individual characteristics of the older adults, the psychological status is directly related to the social cognition and the demand judgment. A positive psychological status will improve the physical health of the older adults, increase the frequency of their interaction with society, and reduce the older adults’ perceived vulnerability, which makes the older adults obtain more healthcare information and increase the potential accessibility. On the contrary, a negative psychological status will not only misjudge the self-demand of healthcare services, but also reduce the willingness to perceive, seek, and approach healthcare services. However, some scholars believe that older adults with negative psychological status have many problems in carrying out activities of daily living. In other words, there is an increased dependency on others and healthcare systems [31], which will increase the opportunity to take advantage of healthcare services. In terms of factors affecting psychological status, they include the change of social roles and responsibility, the economic insecurity due to retirement and the decline of social cognition. Isolation, loneliness, sense of neglect and boredom are the common complaints of the older adults. All these factors influence the psychosocial status of the older adults [32,33], leading to differences of the willingness to use healthcare services.

**Hypothesis** **2** **(H2).**
*The positive psychological status of the older adults can reduce perceived vulnerability and increase potential accessibility.*


The willingness and frequency to use healthcare services are closely related to the income of the older adults. The older adults with higher income have more diversified needs for healthcare services. They pay more attention to the quality and content of healthcare services, which increases the potential accessibility of healthcare services. However, the older adults with lower income are limited by transportation costs, information costs, service costs, etc., and their perceived vulnerability has also increased. They are more willing to use healthcare services nearby, choose some less expensive healthcare services, and reduce using frequency [34]. Seriously, low-income families are more willing to give up the expense on healthcare services [35], thereby reducing the willingness to use healthcare services. So insufficient income will adversely affect health and is directly correlated with inequality in potential accessibility of healthcare service [36]. In addition, expanding income inequality is contributing to decrease potential accessibility of the healthcare service [37].

**Hypothesis** **3** **(H3).**
*Inadequate income of the older adults increases perceived vulnerability and reduces potential accessibility.*


Older people with higher education level have more opportunities to delay retirement or re-employment [38], so the ability to obtain income is stronger, and there are more opportunities to find and access healthcare services. Besides, older people with higher education level have a decrease in attitudes toward filial obligation, and they are more likely to accept diversified healthcare services [39], so the potential accessibility is high. Older adults with lower education level are more willing to rely on family care and less dependent on community healthcare services, so the potential accessibility is low. In addition, lower education level will also affect their health management and disease control, thereby increasing their perceived vulnerability. And individuals are barraged with reading materials when they enter a health care setting. Consent forms, prescriptions, admission paperwork, discharge instructions, advance directives, and educational brochures are some of the things that overwhelm patients with poor education [40], which also reduces the chance of using healthcare services.

**Hypothesis** **4** **(H4).**
*Older people with higher education level have lower perceived vulnerability and higher potential accessibility.*


In rural areas, there are problems such as fragmentary infrastructure and insufficient resources for healthcare services. So, rural areas have problems in attracting and retaining healthcare workforce [41], which leads to poor dependence on healthcare services for rural older adults. The potential accessibility is low. Compared with rural areas, urban older adults have better physical health, life satisfaction and social support [42], so urban older adults have stronger perception to the demand of healthcare services [43]. The potential accessibility of healthcare services is also high. Therefore, the substantial differences associated with economic inequality, social context, social security, and demographic characteristics, cause different willingness to use healthcare services in urban and rural areas.

**Hypothesis** **5** **(H5).**
*The different willingness to use healthcare services in urban and rural areas causes differences on potential accessibility.*


## 3. Data and Index

The data in this article comes from a summer survey of major national projects in 2019. The survey team is composed of more than 20 members of teachers, doctoral students and master students from Xi’an Jiaotong University. This survey is aimed at the accessibility of healthcare services in Shaanxi Province. The survey method adopts stratified sampling. Finally, we investigate three representative regions of Hanzhong, Baoji, and Yan’an, which are characterized by high aging and rapid development of healthcare service for the older adults. The aging characteristics of these areas are similar to the healthcare services in most rural areas in China, and this data can also show some problems of healthcare services in countries that are aging before they get rich. By filtering and selecting the data, we obtain 660 valid questionnaires related to the theme, whose validity value of Cronbach’s α is 0.94, which is representative of the population.

In this article, potential accessibility is used as the dependent variable, distance to service as a spatial factor and perceived vulnerability as a social factor are used as independent variables. At the same time, psychosocial status, income, residence, education level, health status, and gender are used as control variables to ensure that the results are not affected by personal characteristics. The older adults can assess each item according to their actual feelings, using the Likert 5-point method, and assigning values according to “1, 2, 3, 4, 5” from weak to strong accessibility. Descriptive statistics of sample is shown in Table 1.

## 4. Results and Discussion

Based on the above analysis framework, this article uses ordered logit model to verify the above hypotheses, analyzes the empirical results and makes a suggestion. The ordered logit model is a mature statistical method, suitable for correlation analysis between ordered variables.

### 4.1. Perceived Vulnerability

In order to verify Hypothesis 1, this article tests the impacts of distance and perceived vulnerability on potential accessibility, using psychosocial status, income, residence, education level, health status, and gender as control variables. The results are shown in Table 2. It can be seen that the impacts of the distance and perceived vulnerability on potential accessibility are significantly positive, and the results are stable in the conditions of different control variables. The log likelihood test is passed, indicating that the more convenient it is to reach the healthcare services center, the higher the potential accessibility is. The better the perceived ability of the older adults is, the stronger the willingness is to use healthcare services. Hypothesis 1 is supported. According to the results of Model 1–6, the level of potential accessibility is not only affected by distance and perceived vulnerability, but also by individual characteristics such as the psychosocial status, income, residence, education level of the older adults, indicating the differences of the potential accessibility mainly come from the differences of the personal characteristics of the older adults, and different living environments cause differences of the willingness to use healthcare services for the older adults.

On the difference of the potential accessibility of healthcare services, the psychosocial status, income, and educational level of the older adults significantly have a positive influence on the potential accessibility, indicating that the better psychosocial status, higher education level and income, can improve the willingness to use healthcare services, so the development of healthcare services requires financial support and conceptual support. However, residence has a negative impact on potential accessibility, indicating that urban older adults are more willing to use healthcare services. The result shows that urban healthcare services are developing rapidly and have abundant healthcare resources, and the older adults are more likely to use healthcare services.

### 4.2. Psychosocial Status

In order to verify Hypothesis 2, this article tests the impacts of distance and perceived vulnerability on potential accessibility in the conditions of different psychosocial status, using income, residence, education level, health status, and gender as control variables. The results are shown in Table 3.

Perceived vulnerability and distance have greater impacts on potential accessibility for older adults with poor psychosocial status, which are 0.9471 and 0.3446 respectively, indicating that the better the psychosocial status is, the smaller the impact of perceived vulnerability on potential accessibility is. Hypothesis 2 is supported. The mainly causes include three aspects. First, the older adults with better psychosocial status are more confident in their ability to use healthcare services, and it is easy to perceive, find and approach healthcare services. Second, the older adults with better psychosocial status are more willing to participate in social activities and can collect more information on healthcare services, which will reduce the obstacles to find healthcare services. Third, the older adults with better psychosocial status can overcome external obstacles to use healthcare services and increase their potential accessibility.

Although the psychosocial status of the older adults depends on their personal experience and living environment, healthcare service providers can improve the potential accessibility by reducing objective barriers. On one hand, healthcare service providers should widely advertise the information to reduce the information cost of seeking healthcare services. On the other hand, healthcare service centers should reduce the healthcare service fees, and make sure that everyone can use healthcare services facilities and share social service resources. Besides, providers need to expand the coverage of healthcare services, and improve the potential accessibility of healthcare services.

### 4.3. Income

In order to verify Hypothesis 3, this article tests the impacts of distance and perceived vulnerability on potential accessibility in the conditions of different income level, using psychosocial status, residence, education level, health status, and gender as control variables. The results are shown in Table 4.

For the older adults with low income, distance has a significant impact on potential accessibility, indicating that the accessibility of healthcare services is affected by the distance factor. This is because a certain amount of transportation costs are needed to find and approach healthcare services. From the perspective of perceived vulnerability, the willingness of older adults with low incomes to use healthcare services is more likely to be affected by their perceived ability. This is because older adults with low incomes are more likely to doubt their ability to perceive service opportunities, find financial support and use healthcare services. Hypothesis 3 is supported.

In order to increase the opportunities for low-income older adults to use healthcare services, the government should increase the opportunities for reemployment of the older adults and encourage them to increase their income through participation in agricultural production, self-employment, reemployment, and delayed retirement. In addition, the establishment of healthcare service centers should cover a large number of older adults as much as possible to reduce the transportation costs of the older adults. For example, Shaanxi Province established a “15-minute healthcare service circle”.

### 4.4. Education Level

In order to verify Hypothesis 4, this article tests the impacts of distance and perceived vulnerability on potential accessibility in the conditions of different education level, using psychosocial status, income, residence, health status, and gender as control variables. The results are shown in Table 5.

For the older adults with low education level, distance and perceived vulnerability have a significant impact on potential accessibility. This result shows that the willingness of people with low education to use healthcare services is more likely to be affected by distance and perceived ability. Hypothesis 4 is supported. This result can be attributed to three reasons. First, people with low education maybe have lower incomes, so long-distance healthcare services reduce the potential accessibility. Second, people with low education have lower ability to find and use healthcare services. Third, in terms of the older adults with low education level, the inadequate understanding on the relevant policies and information of the healthcare services can easily reduce their enthusiasm for seeking healthcare services.

In order to encourage older adults to actively use healthcare services, firstly, providers should popularize healthcare service knowledge, explain relevant policies for healthcare services, and encourage different types of older adults to use different healthcare services. Secondly, healthcare service providers should increase the content of door-to-door service, which can not only effectively know the real needs of the older adults, but also popularize the knowledge of healthcare services. Finally, we should lower the entry standards for institutions for older adults, popularize the cultural knowledge, increase the opportunities to participate in society, and improve the ability to use healthcare services for the older adults.

### 4.5. Residence

In order to verify Hypothesis 5, this article tests the impacts of distance and perceived vulnerability on potential accessibility in the different residence of urban and rural, using psychosocial status, income, education level, health status, and gender as control variables. The results are shown in Table 6.

Compared with urban areas, the potential accessibility of healthcare services in rural areas is more significantly affected by distance. This is mainly due to the fact that rural older adults are more dispersed, and access to healthcare services requires certain transport costs. Moreover, the perceived vulnerability of rural older adults also has a greater impact on potential accessibility. This is mainly attributed to the limited resources and the lagging information of rural healthcare services. Besides, most of rural older adults rely on family care, and the trust and the consumption awareness in healthcare service centers is low. Hypothesis 5 is supported.

A large number of the older adults will form a huge consumption market of healthcare service in rural areas, so reducing the gap in the potential accessibility between urban and rural areas is conducive to improve the quality of life of the older adults and promote the rapid development of healthcare services. Firstly, we should vigorously publicize the policy of healthcare service, guide the older adults to participate in the healthcare market, and get rid of the traditional concept of children raising the older adults. Secondly, the government needs to increase the income of the rural older adults and reduces the transaction cost to find healthcare services. Finally, the government should introduce a series of subsidy policies on healthcare services in rural, such as increasing resources of healthcare services, cutting down the service fees.

### 4.6. Discussion

According to the above analysis, it can be seen that the potential accessibility of healthcare services is affected by distance and perceived vulnerability. The easier it is for the older adults to reach the healthcare services centers, the higher the potential accessibility is. The poor perceived ability reduces the potential accessibility of healthcare services, and the perceived vulnerability of the older adults depends on individual characteristics such as psychosocial status, income, education level, residence, and gender. This result coincides with the factor of individual characteristic in healthcare services’ use model proposed by Andersen [44]. So, we should reduce the difference on healthcare services to improve the potential accessibility, including insufficient utilization of healthcare resources and perceived vulnerability due to individual differences, which can effectively match the supply and demand of healthcare services. Therefore, the formulation and implementation of healthcare service policies must combine local infrastructure, economic level, consumption habits and cultural customs. On policy publicity, it is necessary to fully consider the education differences of the older adults, concisely introduce the policy content, and timely update the changes in the healthcare service policy. On policy content, it is necessary to explain the contents of the healthcare service in detail, open an information platform, and formulate standards and prices based on the income factors and coverage rate. The healthcare service system should cover all older adults, including disabled older adults, dementia older adults and so on. On policy implementation, we should implement various healthcare services according to the individual characteristic of the older adults. The healthcare services system should change service contents and access way in different regions of urban and rural areas. In short, policies should play a role as a guide, be adapted to local conditions and vary from area to area. We should reduce the perceived vulnerability of the older adults and increase the willingness and opportunities of the older adults to use healthcare services.

Although this article is based on the older adults in Shaanxi Province, the results are applicable to other parts of China. The differences in the psychosocial status, income, education level, residence and other personal characteristics are universal for the older adults in China. The healthcare services in other parts of China have similar problems like Shaanxi. For example, the old-age dependency rate in Shaanxi is 14.99%, which is lower than the Chinese average of 16.77%. Therefore, the problem of healthcare services in other parts of China is more serious. So, those suggestions to improve the potential accessibility are applicable to other parts of China.

## 5. Conclusions

Based on the supply and demand of healthcare services, this article pulls out the spatial and social factors that affect the potential accessibility, and analyzes the reasons of the difference on potential accessibility from the perspective of individual characteristics of psychosocial status, income, education level, residence. This article broadens the theory of potential accessibility from the perspective of perceived vulnerability, and proposes feasible suggestions to improve healthcare services. The main conclusions are as follows.

First, the perceived vulnerability of the older adults reduces the potential accessibility of healthcare services. Most of the older adults have the characteristics of perceived vulnerability on the environment, information, resources and services and so on, so their opportunities and willingness of using healthcare services are low, which reduces the potential accessibility of healthcare services.

Second, the individual characteristics of the older adults cause differences on the potential accessibility of healthcare services. The psychosocial status, income, education level and the willingness to use healthcare services are directly proportional, which promotes potential accessibility. Residence has a negative impact on the willingness to use healthcare services, indicating that the potentially accessible of urban healthcare services is better.

Third, public policies can reduce differences on the potential accessibility of healthcare services. The distribution of healthcare service resources should be allocated according to the potential needs of the older adults, which can improve the matching degree of the supply and demand. The content, publicity and implementation of policies need to consider the individual characteristics and living habits of the local older adults, which can reduce the perceived vulnerability and improve potential accessibility of healthcare services.

## Figures and Tables

**Figure 1 healthcare-08-00452-f001:**
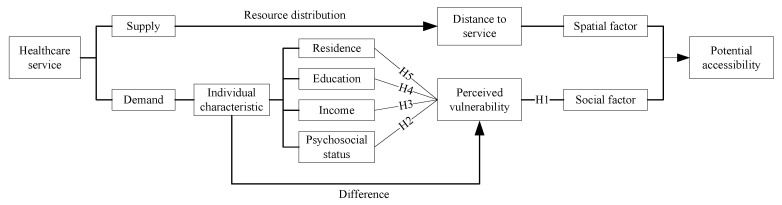
Analysis framework and hypotheses.

**Table 1 healthcare-08-00452-t001:** Descriptive statistics of the study sample.

Variable	Indicator	Item	Mean	Median	Standard Deviation
		(Number)			
Dependent variable	Potential accessibility	Willingness to use healthcare service	3.5957	4	1.1550
Independent variable	Distance to service	Convenience to community healthcare service centers	3.7769	4	1.4304
		(1)			
	Perceived vulnerability	Perceived fairness to use healthcare service	3.4097	3	1.1428
		(2)			
Control variable	Individual characteristics	Psychosocial status	4.2185	4	0.9295
		(3)			
		Income	3.3340	4	0.9557
		(4)			
		Education level	1.9560	2	1.0386
		(5)			
		Residence	1.4249	1	0.4947
		(6)			
		Health status	3.4810	4	1.0835
		(7)			
		Gender	1.6085	2	0.4886
		(8)			

**Table 2 healthcare-08-00452-t002:** The impact of perceived vulnerability on potential accessibility.

Variable	Model1	Model 2	Model 3	Model 4	Model 5	Model 6	Model 7
1	0.2282 ***	0.2760 ***	0.2640 ***	0.2755 ***	0.2373 ***	0.2403 ***	0.2758 ***
	(4.2525)	(5.0992)	(4.9003)	(5.0652)	(4.4394)	4.4997)	(5.0056)
2	0.3699 ***	0.3960 ***	0.3846 ***	0.3943 ***	0.3765 ***	0.3846 ***	0.3814 ***
	(5.5599)	(5.9447)	(5.7907)	(5.9256)	(5.6416)	(5.7848)	(5.6852)
3	0.2185 ***						0.1779 **
	(2.8584)						(2.2191)
4		0.2117 ***					0.1082
		(3.8669)					(1.4836)
5			0.2303 ***				0.1179
			(3.3055)				(1.4388)
6				-0.4641***			−0.2065
				(-3.1631)			(−1.1611)
7					0.1026		−0.003
					(1.5464)		(−0.0428)
8						−0.0979	0.0339
						(−0.6760)	(0.2245)
Log likelihood	−926.90	−923.43	−925.46	−925.96	−929.78	−930.75	−918.85

Note: T-values are in parentheses. *** means significant at the 1% level, ** means significant at the 5% level, and * means significant at the 10% level.

**Table 3 healthcare-08-00452-t003:** The impact of psychosocial status on potential accessibility.

Variable	Potential Accessibility	Potential Accessibility
	(Poor Psychosocial Status)	(Better Psychosocial Status)
1	0.3446 ***	0.2596 ***
	(2.5286)	(4.2761)
2	0.9471 ***	0.2731 ***
	(5.3821)	(3.6968)
4	0.3237	0.0940
	(1.6350)	(1.1789)
5	0.3396	0.0853
	(1.3622)	(0.9762)
6	−0.4743	−0.1510
	(−1.0168)	(−0.7743)
7	−0.3721	0.0654
	(−2.2842)	(0.8217)
8	−0.0950	0.0747
	(−0.2218)	(0.4588)
Log likelihood	−153.58	−754.85

Note: T-values are in parentheses. *** means significant at the 1% level, ** means significant at the 5% level, and * means significant at the 10% level.

**Table 4 healthcare-08-00452-t004:** The impact of income on potential accessibility.

Variable	Potential Accessibility	Potential Accessibility
	(Low Income)	(High Income)
1	0.3963 ***	0.0972
	(5.4611)	(1.1221)
2	0.4339 ***	0.2916 ***
	(5.0954)	(2.5817)
3	0.1743 *	0.2629 *
	(1.7919)	(1.8004)
5	0.3396	0.0543
	(1.3622)	(0.4802)
6	0.2330 **	−0.6045
	(2.0020)	(−1.4474)
7	−0.0696	0.1134
	(−0.7996)	(0.8837)
8	−0.3082	0.6691 ***
	(−1.5855)	(2.6762)
Log likelihood	−589.77	−315.03

Note: T-values are in parentheses. *** means significant at the 1% level, ** means significant at the 5% level, and * means significant at the 10% level.

**Table 5 healthcare-08-00452-t005:** The impact of education level on potential accessibility.

Variable	Potential Accessibility	Potential Accessibility
	(Low Education Level)	(High Education Level)
1	0.2888 ***	0.2396
	(4.9778)	(1.2526)
2	0.3745 ***	0.2799
	(5.3846)	(0.9361)
3	0.1718 **	0.2422
	(2.0541)	(0.8214)
4	0.1674 **	−0.3996
	(2.3318)	(−1.1548)
6	−0.1784	−1.2266
	(−0.9812)	(−1.3723)
7	−0.1180	0.0021
	(−0.1590)	(0.0077)
8	−0.1110	1.2324 **
	(−1.7178)	(2.1312)
Log likelihood	−848.01	−64.19

Note: T-values are in parentheses. *** means significant at the 1% level, ** means significant at the 5% level, and * means significant at the 10% level.

**Table 6 healthcare-08-00452-t006:** The impact of residence on potential accessibility.

Variable	Potential Accessibility	Potential Accessibility
	(Urban)	(Rural)
1	0.1774 ***	0.5008 ***
	(2.7136)	(4.9050)
2	0.3682 ***	0.4090 ***
	(4.2470)	(3.7999)
3	0.1551	0.2413 *
	(1.4340)	(1.9526)
4	0.0969	0.0938
	(1.0024)	(0.8232)
5	0.1971 **	−0.0434
	(1.9803)	(−0.2893)
7	0.0923	−0.1264
	(0.9503)	(−1.1759)
8	0.3803 *	−0.4522
	(1.9092)	(−1.9129)
Log likelihood	−517.30	−392.03

Note: T-values are in parentheses. *** means significant at the 1% level, ** means significant at the 5% level, and * means significant at the 10% level.

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
