# Peer review of "Differences in the Potential Accessibility of Home-Based Healthcare Services among Different Groups of Older Adults: A Case from Shaanxi Province, China"

_healthcare, 2020, doi:10.3390/healthcare8040452_

Round 1
Reviewer 1 Report
The manuscript provides interesting results that could improve the accessibility of the elderly to health services. However, I think that in order to be published, it should improve some aspects:
- On line 30, please clarify what you mean by "spiritual comfort".
In the Introduction section, they refer to economic factors as elements that hinder access to health services. I consider it necessary to include a paragraph that helps readers to contextualize the type of health services that exist in the area where the study is carried out. If they are public or private, some reference to help identify the economic cost of accessing them in relation to the average income of the elderly in the area, etc. In section 3 "Data and Index" they offer a brief explanation about the context of the geographic areas of the study, but I believe that this information should be expanded to help the reader determine the level of generalization or transfer of the study results to other areas of other countries. - On line 63, please clarify what you mean by "collaborative leadership initiatives", expanding on this information or giving suitable examples.
- In lines 185-189, I do not consider it necessary to use the bold type format.
- The section "4.6 Discussion" should be greatly improved. The authors do not discuss their results with internationally relevant studies. They must do so in order to give consistency to the discussion.
- At line 325, the authors state that these results are applicable to other parts of China. If this is the case, they should provide adequate data and references to support these claims, as well as the rest of the study discussion.
Author Response
Point 1: On line 30, please clarify what you mean by "spiritual comfort".
Response 1: Replace "spiritual comfort with" with "mental care" on line 31.
Point 2: In the Introduction section, they refer to economic factors as elements that hinder access to health services. I consider it necessary to include a paragraph that helps readers to contextualize the type of health services that exist in the area where the study is carried out. If they are public or private, some reference to help identify the economic cost of accessing them in relation to the average income of the elderly in the area, etc.
Response 2: In my opinion, though the economic factors are important, this article focus on two factors of the perceived vulnerability and distance. In terms of potential accessibility, we emphasize the opportunity and willingness of users to the service system. The economic factors directly affect the realized accessibility, which is next topic that we will study.
Point 3: In section 3 "Data and Index" they offer a brief explanation about the context of the geographic areas of the study, but I believe that this information should be expanded to help the reader determine the level of generalization or transfer of the study results to other areas of other countries.
Response 3: Increase more information that the aging characteristics of these areas are similar to the healthcare services in most rural areas in China, and this data can also show some problems of healthcare services in countries that are aging before they get rich on line 191-193.
Point 4: On line 63, please clarify what you mean by "collaborative leadership initiatives", expanding on this information or giving suitable examples.
Response 4: Delete the word "initiatives" that is a redundant word on line 67.
Point 5: The section "4.6 Discussion" should be greatly improved. The authors do not discuss their results with internationally relevant studies. They must do so in order to give consistency to the discussion.
Response 5: Increase a discussion. This result coincides with the factor of individual characteristic in healthcare services' use model proposed by Andersen [44] on 332-333.
Point 6:At line 325, the authors state that these results are applicable to other parts of China. If this is the case, they should provide adequate data and references to support these claims, as well as the rest of the study discussion.
Response 6: Increase adequate data. For example, the old-age dependency ratio in Shaanxi is 14.99, which is lower than the Chinese average of 16.77. Therefore, the problem of elderly care services in other parts of China is more serious on line 353-355.
More details are in attachment.

Reviewer 2 Report
I would like to thank Authors for important and well written manuscript about actual topic- potential accessibility of home based healthcare services- where example from Shaanxi Province in China has been explored.
I have two minor comments regarding content of manuscript; if authors would agree on minor clarifications in the manuscript it would contribute to more precise presentation of information about this study.
1) Analysis framework
Authors explained five hypothesis in text and indicate them by abbreviation (H). Even thought that it is well known abbreviation used in scientific publications, I would recommend to indicate that in text, eg. in line 104: The analysis framework and hypothesis (H) are shown...
2) Data and Index
The data for this study comes from survey performed in the national project. It would be helpful for reader to have some insight about nature of questions included in the survey, particularly those which are used as variables in current study. Such information helps readers to better understand the study procedure and contextualize the findings.
In case if data collection is described elsewhere, the reference would be enough.
Author Response
Point 1: Authors explained five hypothesis in text and indicate them by abbreviation (H). Even thought that it is well known abbreviation used in scientific publications, I would recommend to indicate that in text, eg. in line 104: The analysis framework and hypothesis (H) are shown...
Response 1: Make a revision. The analysis framework and hypothesis (H) are shown... on line 109.
Point 2:The data for this study comes from survey performed in the national project. It would be helpful for reader to have some insight about nature of questions included in the survey, particularly those which are used as variables in current study. Such information helps readers to better understand the study procedure and contextualize the findings.
Response 2: Increase more information. The older adults can score each item according to their actual feelings, using the Likert 5-point scale method, and assigning values according to "1, 2, 3, 4, 5" from weak to strong accessibility on line 199-201.
More details are in attachment.

Reviewer 3 Report
Using the word elderly or old people is not appropriate. I suggest use the word Older Adults or Geriatric Age group people.
Unbalanced is wrong at 2 places, it should be imbalanced
Can explain research methods precisely.
The results section could be elaborated and tabulated more precisely.
It’s a good study but it would be wiser to improve on the language and edit results to make it more attractive to read and understand.

Author Response
Point 1: Using the word elderly or old people is not appropriate. I suggest use the word Older Adults or Geriatric Age group people. Unbalanced is wrong at 2 places, it should be imbalanced.
Response 1: Replace the word elderly or old people with the word older adults. Replace unbalanced with imbalanced on line 14 and 41.
Point 2: Can explain research methods precisely.
Response 2: Increase an explanation. The ordered logit model is a mature statistical method, suitable for correlation analysis between ordered variables on line 206-208.
Point 3:The results section could be elaborated and tabulated more precisely.
Response 3: Increase some data on line 353-355, and increase some discussions with internationally relevant studies on line 333-334.
More details are in attachment.
